# The Usefulness of Optical Coherence Tomography in Disease Progression Monitoring in Younger Patients with Relapsing-Remitting Multiple Sclerosis: A Single-Centre Study

**DOI:** 10.3390/jcm12010093

**Published:** 2022-12-22

**Authors:** Magdalena Torbus, Ewa Niewiadomska, Paweł Dobrakowski, Ewa Papuć, Barbara Rybus-Kalinowska, Patryk Szlacheta, Ilona Korzonek-Szlacheta, Katarzyna Kubicka-Bączyk, Beata Łabuz-Roszak

**Affiliations:** 1Institute of Psychology, Humanitas University in Sosnowiec, 41-200 Sosnowiec, Poland; 2Department of Biostatistics, Faculty of Health Sciences in Bytom, Medical University of Silesia, 40-055 Katowice, Poland; 3Department of Neurology, Medical University of Lublin, 20-093 Lublin, Poland; 4Department of Basic Medical Sciences, Faculty of Health Sciences in Bytom, Medical University of Silesia, 40-055 Katowice, Poland; 5Department of Toxicology and Health Protection, Faculty of Health Sciences in Bytom, Medical University of Silesia, 40-055 Katowice, Poland; 6Department of Prevention of Metabolic Diseases, Faculty of Health Sciences in Bytom, Medical University of Silesia, 40-055 Katowice, Poland; 7Department of Neurology, Faculty of Medical Sciences in Zabrze, Medical University of Silesia, 40-055 Katowice, Poland; 8Department of Neurology, Institute of Medical Sciences, University of Opole, 45-040 Opole, Poland

**Keywords:** multiple sclerosis, optical coherence tomography, neurodegeneration, cognitive impairment, disease monitoring

## Abstract

The purpose of the study was to assess the usefulness of optical coherence tomography (OCT) in the detection of the neurodegenerative process in younger patients with multiple sclerosis (MS). The study group consisted of 61 patients with a relapsing remitting course of MS (mean age 36.4 ± 6.7 years) divided into two groups: short (≤5 years) and long (>10 years) disease duration. OCT, P300 evoked potential, Montreal Cognitive Assessment, and performance subtests (Picture Completion and Digit Symbol) of the Wechsler Adult Intelligence Scale were performed in all patients. Mean values of most parameters assessed in OCT (pRNFL Total, pRNFL Inferior, pRNFL Superior, pRNFL Temporalis, mRNFL, GCIPL, mRNFL+GCIPL) were significantly lower in MS patients in comparison to controls. And in patients with longer disease duration in comparison to those with shorter. Most OCT parameters negatively correlated with the EDSS score (*p* < 0.05). No significant correlation was found between OCT results and both P300 latency and the results of psychometric tests. OCT, as a simple, non-invasive, quick, and inexpensive method, could be useful for monitoring the progression of disease in MS patients.

## 1. Introduction

Multiple sclerosis (MS) is a complex neurodegenerative disorder of the brain and spinal cord that is associated with autoimmune mechanisms involving white and grey matter. It is characterized by progressive demyelination, gliosis, axonal dysfunction as well as loss of neurons. Symptoms of MS vary depending on the location of the lesions in the central nervous system (CNS). These are visual disturbances, muscle weakness, bulbar and oculomotor symptoms, sensory disturbances, and symptoms of autonomic dysfunction. Cognitive impairment is also a significant problem, which may affect 40–75% of patients [1,2,3].

One of the non-invasive methods that can help in the assessment of neuronal and axonal damage is optical coherence tomography (OCT) of the retina.

The retina and the optic nerve are part of the CNS. There are three types of neurons: photoreceptors, bipolar cells, and ganglion cells that make up the visual pathway. Amacrine cells and horizontal cells are located between them. They modulate signals of intercellular transmission [4,5]. The retinal nerve fiber layer (RNFL) consists of unmyelinated ganglion cell axons. They are covered with myelin only when they leave the eyeball, forming the optic nerve. The translucency of the eyeball makes it possible to visualize the retina as a “window” through which one can look into CNS [6,7,8]. Lack of myelin and low concentration of glial cells in the retina create ideal conditions to assess the pathophysiology and development of neurodegenerative diseases.

Post-mortem analysis of MS patients showed damage to the optic nerves in 94–99%, even if they did not have a clinical episode of optic neuritis (ON) [9]. Axonal demyelination of the optic nerve leads to primary axonal thinning and secondary degeneration of the optic nerve with further degeneration of the ganglion cells. This is reflected in OCT by thinning of RNFL and macular ganglion cell layer (GCL) [10], the ethology of axonal atrophy and loss of GCL in MS without ON is unknown. Most likely it may be related to retrograde transsynaptic degeneration of retinal neurons. Researchers observed thinning of RNFL in all quadrants or only in some quadrants (most often temporal, more rarely lower and upper), more severe in progressive forms of MS (secondary progressive, SPMS; and primary progressive, PPMS) when compared to relapsing-remitting MS (RRMS) and clinically isolated syndrome (CIS) [11,12,13,14,15].

Studies have shown that assessing GCL and the inner plexiform layer (IPL) can more sensitively predict axonal damage than assessing RNFL alone [16]. The degree of atrophy corresponds to the loss of the cortex volume. Britze et al. in their meta-analysis [16] showed that GCL+ (Ganglion Cell + Inner Plexiform Layer; a parameter that combines GCL and IPL) was significantly reduced in MS patients with a previous episode of ON. However, without the history of ON, this parameter was also decreased. Britze et al. [16] observed a correlation between the reduction of GCL+ and the increase in disability of patients assessed by the Expanded Disability Status Scale (EDSS). This suggests that GCL+ may be a good marker of neurodegeneration.

The early detection of commencing neurodegenerative processes at the subclinical level may help in selecting patients at particular risk of developing cognitive disorders and enable early implementation of appropriate preventive measures.

The aim of this study was to assess the degree of axonal and neuronal damage in younger RRMS patients with different disease duration using OCT of the retina, and to determine its usefulness in the early detection of commencing the neurodegenerative process. Moreover, the aim of the study was to evaluate the association between OCT parameters and clinical features of the disease, neurophysiological and psychometric tests.

## 2. Materials and Methods

The study was conducted between July 2015 and April 2018. It was approved by the Institutional Bioethical Board of the Medical University of Silesia in Katowice and was conducted in accordance with the Declaration of Helsinki. Written informed consent was obtained from all study participants.

Two cohorts were recruited (*N* = 82): an MS cohort and a healthy control subjects cohort. Subjects with RRMS (relapsing remitting multiple sclerosis) confirmed by a treating neurologist, according to the current McDonald criteria, were recruited from patients with a relapsing remitting form of the disease, at the age ≥ 18 and ≤45 years, with a duration of the disease (time from onset of first symptoms) ≤ 5 years (group 1a) or ≥10 years (group 1b), EDSS ≤ 4.5 points, treated at the Department of Neurology and Outpatient Clinic in Zabrze, Medical University of Silesia, Poland.

Patients with optic neuritis (ON) in the last 6 months or other relapses in the last month prior to the recruitment, with a history of other than MS accompanying neurological diseases, chronic internal diseases, or ophthalmological disorders (like opacities in the optic centres of the eyeball, retinal disorders, optic disc abnormalities, visual defect greater than (+/−) 4.0 D Sph, (+/−) 2.0 D Cyl) were excluded from the study.

The control group consisted of age and sex-matched healthy volunteers recruited from the staff of the Department of Neurology.

A detailed interview was conducted with respondents using an original questionnaire. The questions concerned age, education, the course of the disease, and the treatment used. The medical documentation was also thoroughly analyzed. A neurological examination of all the patients was performed by one experienced neurologist. The Expended Disability Status Scale (EDSS) score and Functional Systems Scores (FSS) were also determined for each MS patient participating in the study [17].

A full ophthalmological examination was performed by an experienced ophthalmologist. The eye optical system refraction, distance, and near visual acuity tests without correction and with the best ocular correction, evaluation in a slit lamp, and measurement of intraocular pressure with a non-contact tonometer were performed. Then, optical coherence tomography (OCT) was performed with the Topcon 3D OCT-2000 FA plus apparatus using optic disc and macula evaluation protocols. Both of the patient’s eyes were examined. The following parameters were assessed: pRNFL (thickness of peripapillary retinal nerve fiber layer, assessed in total and in 4 quadrants: inferior, superior, temporalis and nasal), mRNFL (thickness of macular retinal nerve fiber layer), GCIPL (thickness of macular ganglion cell—GCL and inner plexiform layer–IPL; a parameter that combines GCL and IPL), mRNFL+GCIPL (a parameter that combines the value of macular RNFL and GCIPL).

The free Polish version of the Montreal Cognitive Assessment (MoCA) and two performance subtests (Picture Completion and Digit Symbol) of the Polish version of the Wechsler Adult Intelligence Scale (WAIS-R) were used to assess cognitive functions. The examination was performed by an experienced psychologist.

The total possible score in MoCA is 30 points; a score of 26 or above is considered normal. It assesses different cognitive domains: attention and concentration (6 points; MoCA Attention), naming abilities (3 points; MoCA Naming), short-term memory (5 points; MoCA Memory), language (3 points; MoCA Language), visuospatial and executive skills (5 points; MoCA Visuospatial), conceptual and abstractive thinking (2 points; MoCA Abstraction), and orientation (6 points; MoCA Orientation) [18].

The Digit Symbol Subtest uses numbers and symbols to replace them; the person is asked to replace a series of digits with those symbols. This test measures the ability to learn new skills, concentration, short-term memory, and hand-eye coordination. To perform the task well, the patient must involve visual, motor and learning activities at the same time. The result (in points) depends on the number of valid elements in 120 s [19].

The Picture Completion Subtest consists of pictures in which important details are missing, e.g., the lack of a handle on a door drawing. The task is to detect and name this detail. This test measures the subject’s perceptive and conceptual abilities, attention and level of knowledge. Thus, this test examines contact with reality, sensitivity to detail, awareness of environmental elements, visual perception, visual long-term memory, and speed of perception. The result (in points) depends on the speed of execution and the number of details found [19].

The neurophysiological study—endogenous P300 evoked potential—was assessed with the Neuro-MEP-4 apparatus from Neurosoft^®^ using surface plate electrodes. The auditory stimulus was used—the subject’s task was to count significant, distinguished stimuli (65 dB nHL, 2000 Hz), randomly distributed among neutral stimuli with different parameters (65 dB nHL, 1000 Hz). The latency of the P300 potential was calculated as the mean value of 50 measurements. The result was expressed in milliseconds (ms). The test was performed at an ambient temperature of 21–24 °C, after rest, about 3–4 h after a meal.

Statistical analysis was made using the R 3.1.2 statistical package under the GNU GPL license and the STATISTICA v.12 program, Stat Soft Poland. Measurable data were presented in the form of mean X and standard deviation S as well as median and interquartile range IQR. The compliance of the distribution of variables with the normal distribution was checked using the Shapiro-Wilk test. In the case of two groups, the significance of mean differences was checked with the Student’s *t*-test, and for three or more groups—the ANOVA test. When the distributions of the variables were asymmetric, the U Mann-Whitney and Kruskal–Wallis tests were used. Multiple comparisons were performed based on the results of post-hoc tests for ANOVA and the Kruskal–Wallis test. Additionally, a parametric or non-parametric two-way analysis of variance was used. The multiple backward stepwise regression analysis was used to confirm a relationship between OCT parameters and particular variables: patients’ age, sex, education, disease duration, EPSS score, number of relapses. We checked the collinearity of independent variables used in the model for excluding redundant variables in the models. Percentage notation was used for nominal data. The differences between the fractions were assessed with the use of the significance test of percentage differences and the Bonferonni test of the significance of percentage differences for more than two groups. The occurrence of relationships between nominal variables was verified with the χ2 test or Fisher’s exact test. The dependence of measurable variables was assessed by determining the Spearman correlation coefficient R and the significance test. The *p*-value < 0.05 was adopted as the criterion of statistical significance.

The relationships between OCT parameters and gender, age, education, age of first symptoms, duration of the disease, EDSS, and the used disease modifying drug (DMD) were analyzed.

## 3. Results

A total of 61 patients with RRMS aged 21 to 45 years were involved in the study (mean 36.4 ± 6.7 years). Among them, there were 42 women and 19 men, divided into two groups:MS I (1a): 30 patients with short duration of the disease (time from first symptoms to time of research) ≤ 5 years (mean 2.77 ± 1.5 years). This group included 21 women (70%) and 9 men (30%), aged 21 to 45 (mean age 33.7 ± 6.9 years)MS II (1b): 31 patients with a long duration of the disease > 10 years (mean 13.16 ± 4 years). This group consisted of 21 women (68%) and 10 men (32%), aged 26 to 45 (mean age 39.1 ± 5.4 years).

The control group consisted of 21 healthy people, aged 19 to 45 (mean 33.9 ± 7.4 years), 16 women (76%), and 5 men (24%). There was no statistically significant difference between the studied groups in terms of gender, while MS II patients were significantly older than MS I patients (*p* < 0.05). No statistically significant differences were also found between investigated groups in terms of sociodemographic factors (education) (Table 1).

Clinical characteristics (age of first symptoms, age of diagnosis, disease duration, number of relapses, EDSS, FSS) of investigated subjects are presented in Table 2. The patients were treated with interferon beta (*n* = 28; 16 in 1a and 12 in 1b), glatiramer acetate (*n* = 3; 2 in 1a and 1 in 1b), fingolimod (*n* = 8; 4 in 1a and 4 in 1b) and natalizumab (*n* = 8; 1 in 1a and 7 in 1b). However, 14 patients (7 in 1a and 7 in 1b) did not receive disease-modifying drugs (DMD). 35 patients (15 in 1a and 20 in 1b) had a history of unilateral optic neuritis (nobody experienced bilateral optic neuritis).

Mean values of most parameters assessed in OCT (pRNFL Total, pRNFL Inferior, pRNFL Superior, pRNFL Temporalis, mRNFL, GCIPL, mRNFL+GCIPL) were significantly lower in MS patients in comparison to the control group (Table 3). Moreover, when comparing groups of patients with different disease duration, it was shown that in group 1b (with longer disease duration) those values (except for pRFNL Nasal, pRFNL Superior) were significantly lower when compared to group 1a. However, in MS patients with shorter disease duration (≤5 years), the values of OCT parameters were lower compared to the control group, but without statistical significance (Table 3).

The results of psychometric tests are presented in Table 4. The mean MoCA score was statistically significantly lower in MS patients compared to the control group (26.8 ± 2.9 and 29.5 ± 0.9, respectively; *p* < 0.0001), as well as in patients with longer disease duration (1b), in comparison to patients with shorter disease duration (1a) (25.9 ± 2.9 and 27.7 ± 2.6, respectively; *p* < 0.05). The analysis of individual domains showed that MS patients obtained significantly lower results in MoCA Visuospatial, MoCA Attention, and MoCA Memory in comparison to the control group.

MS patients achieved statistically significantly worse results than healthy volunteers in the Picture Completion Subtest (27.4 ± 4.4 vs. 31 ± 3 points; *p* < 0.001). Moreover, the results of MS patients with longer disease duration (1b) were significantly worse than those of subjects with shorter disease duration (1a) (25.9 ± 4.9 vs. 28.9 ± 3.4 points; *p* < 0.05).

The results of the Digit Symbol Test were similar in MS and control groups (52.6 ± 12.7 vs. 56.4 ± 7.1 points). However, when subgroups were analyzed, significantly worse results were achieved by MS patients with longer disease duration (1b) (44.5 ± 10 points) when compared to those with shorter disease duration (1a) (61 ± 9.3 points; *p* < 0.0001).

P300 latency was shown to be prolonged in MS patients (318.1 ± 28.3 ms) compared to the control group (297.3 ± 16.2 ms; *p* < 0.01). Although no statistically significant differences were found between the two groups of patients with MS, an increase in P300 latency was observed in subjects with longer disease duration (MS II) compared to those with shorter disease duration (MS I) (323.5 ± 27.5 vs. 312.6 ± 28.6 ms) (Table 4).

The results of the extended analysis are presented in Table 5, Table 6, Table 7, Table 8 and Table 9.

No statistically significant correlation was found between the OCT parameters and the results of psychometric tests (MoCA: Visuospatial, Attention, Language, Orientation, Digit Symbol, Picture Completion) either in the group of all MS patients or in patients with shorter (1a) and longer (1b) disease duration. Also, no statistically significant correlation was found between OCT results and P300 latency either in the whole group of MS patients or in subgroups (1a and 1b) (Table 5). A significant (*p* < 0.05) negative correlation was found between the P300 latency and the results of the Picture Completion Subtest (R = −0.33), MoCA Total (R = −0.26) and MoCA Abstraction (R = −0.27) in the whole group of MS patients.

There were no statistically significant differences between men and women with regard to the parameters tested in OCT. Significantly lower values of pRNFL Total, pRNFL Inferior, pRNFL Temporalis, mRNFL, GCIPL, and mRNFL+GCIPL were present in older MS patients (>40 years) compared to younger ones. As for the education level, significantly lower values of OCT parameters (pRNFL Total) were observed in MS patients with primary education in relation to patients with secondary and higher education (Table 6).

Among patients with first MS symptoms occurring below 30 years, most of examined OCT parameters (only except for pRNFL Nasal) were statistically significantly lower in patients with longer MS duration (>10 years) in relation to people with shorter MS duration (≤5 years). On the other hand, in patients with the first symptoms of MS over 30 years, there were no statistically significant differences between subjects with longer and shorter disease duration. The use of DMD had no significant effect on the results of OCT (Table 7).

Most of the OCT parameters (pRNFL Total, pRNFL Inferior, pRNFL Temporalis, mRNFL, GCIPL, GCIPL+mRNFL) were negatively correlated with EDSS score in the group of all MS patients. A strong relationship was observed in group 1b between EDSS and mRNFL, GCIPL and GCIPL+pRNFL (Table 8). A negative correlation was found between the number of relapses and GCIPL and GCIPL+mRNFL. There was also a statistically significant relationship between longer disease duration (time from first symptoms) and lower values of the following OCT parameters: pRNFL Total, pRNFL Inferior, pRNFL Nasal, pRNFL Temporalis, mRNFL, GCIPL, mRNFL+GCIPL (*p* < 0.001) (Table 8). Additional multiple regression analysis confirmed a significant influence of patients’ age in the case of pRNFL Total, pRNFL Inferior, pRNFL Temporalis, and MS duration time in: pRNFL Total, mRNFL, GCIPL, GCIPL + mRNFL (Table 8).

Table 9 presents the relationship between functional systems scores and OCT parameters. In the group of all MS patients, we found a statistically significant negative correlation between brainstem function score and pRNFL Total (*p* < 0.01), pRNFL Inferior (*p* < 0.01), pRNFL Temporalis (*p* < 0.05), and mRNFL (*p* < 0.05). For the pyramidal system score, a significant negative correlation was demonstrated with mRNFL (*p* < 0.01), GCIPL (*p* < 0.05), and mRNFL+GCIPL (*p* < 0.05); for the cerebellar system score—a significant negative correlation with the pRNFL Total (*p* < 0.05), pRNFL Temporalis (*p* < 0.05), pRNFL Superior (*p* < 0.05), mRNFL (*p* < 0.05), GCIPL (*p* < 0.01), mRNFL+GCIPL (*p* < 0.01); for bladder and bowel function scores—a significant negative correlation with pRNFL Total (*p* < 0.001), pRNFL Inferior (*p* < 0.001), pRNFL Nasal (*p* < 0.05), pRNFL Superior (*p* < 0.05), mRNFL (*p* < 0.01), GCIPL (*p* < 0.01), mRNFL+GCIP (*p* < 0.01). Moreover, sensory system function score correlated statistically significantly negatively with pRNFL Temporalis (*p* < 0.05), mRNFL (*p* < 0.05), and mRNFL+GCIPL (*p* < 0.05). There was no significant correlation between the parameters assessed in OCT and scores for the assessment of visual functions, and ambulation index (Table 9).

## 4. Discussion

Parisi et al. were the first who used OCT to examine MS patients with a history of optic neuritis [20]. They reported a significant reduction in the thickness of RNFL in eyes with ON compared to the control group and in the affected eye compared to the healthy eye in the same patient [20]. Recurrent ON significantly reduces the RNFL value as compared to eyes with a history of only one episode of ON [21]. Balcer et al. [22] showed a difference between the thickness of RNFL in MS patients (95.5 µm ± 14.5 µm) compared to the control group (104.5 µm ± 10.7 µm). It should be remembered that during the acute phase of ON, the thickness of RNFL may increase in 82% of cases [23]. This is most likely due to the inflammation and swelling of the macula. It is only after a few months that a decline in the RNFL can be noticed.

The thickness of RNFL is decreased not only in the eyes with a history of previous ON, but also without such a history, which may be a direct marker of the progression of brain atrophy in MS patients. In OCT one can also examine the macular area that contains the ganglion cells, therefore GCIPL parameter can be an indicator of axonal damage. We examined a total of 164 eyes—122 eyes of MS patients and 42 eyes in the control group. Except for Nasal RNFL, the mean values of OCT parameters were significantly different compared to the control group. The study by Garcia-Martin et al. [24] demonstrated thinning of RNFL in all quadrants. Moreover, after 5 years of follow-up, researchers demonstrated a reduction in RNFL in all quadrants, as well as in GCIPL and macular volume, both in the MS group and in the control group, but patients with MS had a significantly greater loss. Moreover, according to this study, a history of ON had no effect on the final result.

Many researchers have shown a close relationship between OCT results (mainly RNFL and GCIPL thickness) and some MR measures (like normalized brain parenchymal volume, cortical grey matter volume, normalized white matter volume, number and volume of cortical lesions, thalamus volume, insula volume) [25,26,27,28,29,30]. Therefore, in this study, OCT was used to evaluate neurodegeneration in young patients with multiple sclerosis.

In this study, we confirmed the progression of the neurodegenerative process in MS subjects with longer disease duration. Patients suffering from MS for 10 years or more had significantly lower OCT parameters than those suffering from MS for no longer than 5 years. Interestingly, patients with shorter disease duration (up to 5 years), had worse OCT results than healthy people, but the difference was not statistically significant.

The results of our study also confirm the usefulness of OCT in assessing the risk of disability progression. Similar to Garcia-Martin et al. [24], we confirmed a strong correlation between most OCT parameters and the EDSS score. Additionally, the present study revealed a correlation between most OCT parameters and some functional system scores (pyramidal, brainstem, cerebellum, sensory, bowel, and bladder). The multicentre study by Martinez-Lapiscina et al. [31] confirmed the use of RNFL as a marker of disability progression in MS. This finding could make it much easier for neurologists to monitor disease progression, and—what is more important—help in making treatment decisions based on OCT as a marker of the neurodegenerative process.

In our study MoCA and two performance subtests (Picture Completion and Digit Symbol) of Polish version of WAIS-R were used to assess cognitive functions. MS patients achieved statistically significantly worse results than healthy volunteers in the MoCA test and in the Picture Completion Subtest. Moreover, the results of all tests were significantly worse in patients with longer disease duration.

The data on the relationship between psychometric tests and OCT results are equivocal. El Ayoubi et al. found correlations between the annualized change in GCIPL thickness and annualized change in MoCA scores among MS patients [32]. Also, Esmael at al. observed a significant positive correlation between the cognitive functioning of MS subjects evaluated by the MoCA test and OCT parameters (RNFL and GCIPL) [33].

Contrary to the above results, Oktem et al. [34] have not found a statistically significant association between the thickness of retinal layers and MoCA results, but among patients with Alzheimer’s disease and MCI. Similarly to this study, we also found no correlation between selected psychometric tests (MoCA and two subtests of WAIS) and OCT parameters.

Similar to other studies, P300 latency was prolonged in MS patients compared to the control group (*p* < 0.01) [35,36]. We found no statistically significant correlation between OCT results and P300 latency both in the whole group of our MS patients and in investigated subgroups. The relationship between the OCT and P300 latency has not been assessed in previously performed studies, so further studies on a larger group of patients would be relevant.

In our study, the use of DMDs had no significant effect on the results of OCT. However, the subgroups of patients treated with each drug were small and it could have influenced this observation. Current DMTs are effective in inhibiting neuroinflammation and reducing the rate of clinical relapses but their neuroprotective effects are less well-defined. Data on the relationship between OCT results and DMDs are unclear. Knier et al. analyzed the association between OCT changes and DMT treatment in patients with MS [37]. The authors did not find any difference between patients with or without DMDs but they examined only subjects with first-line drugs (interferon beta, glatiramer acetate, dimethyl fumarate, and teriflunomide). On the other hand, You at al. observed that progressive loss of retinal ganglion cells was more pronounced in patients treated with interferon beta than other DMTs (glatiramer acetate, fingolimod, dimethyl fumarate, natalizumab, alemtuzumab, rituximab, and ocrelizumab) [38]. Zivadinov et al. noticed that glatiramer acetate might have a neuroprotective effect against RNFL loss [39]. Button et al. found that GCIPL thinning in patients on both interferon beta and glatiramer acetate is faster than in patients on natalizumab [40].

Our study confirmed that neuronal and axonal damage in MS progresses with the duration of the disease, which was reflected by thinning of RNFL and GCIPL. These two parameters correlated positively with the degree of the neurodegenerative process. It is also worth noting that the inclusion of healthy control subjects in our study was justified as they had significantly higher values of investigated OCT parameters.

Our study also confirms that the neurodegenerative process in the investigated group of MS patients was associated with the progression of the disease, and not, for example, with the age of the patients.

It is worth emphasizing that, according to our knowledge, this is the only study in which MS patients were simultaneously evaluated by three different methods: OCT, neurophysiological (P300), and psychometric tests (MoCA, two performance subtests of the Polish version of WAIS-R—Picture Completion and Digit Symbol).

OCT, as a simple, non-invasive, quick and inexpensive method, could be useful for monitoring the progression of neurodegeneration in MS, and thus identify patients at particular risk of developing cognitive disorders, which allows for quick implementation of preventive actions, especially cognitive rehabilitation.

## 5. Conclusions

OCT, as a simple, non-invasive, quick and inexpensive method, could be useful for monitoring progression of disease in MS patients.

## 6. Limitations

Our study has some limitations. Firstly, the study group was small and came only from one centre. Secondly, although some researchers believe [31,41] that the analysis of OCT parameters in eyes without a history of ON is more reliable to illustrate the processes of neurodegeneration than in eyes with a history of ON, we assessed both eyes of each patient regardless of the history of ON, and data from eyes affected by ON and data from eyes not affected by ON were given together. However, patients with ON in the last 6 months prior to the recruitment were excluded from the study. Thirdly, we did not evaluate the inner nuclear layer (INL), the parameter that recently has been discussed as a marker of neurodegeneration in MS [42,43,44,45], so further study is needed in this field. Finally, we compared the results from people with longer and shorter disease duration, but we did not perform follow-up examinations, and it would be interesting to see how the OCT results changed over time as the disease progressed.

## Figures and Tables

**Table 1 jcm-12-00093-t001:** Sociodemographic characteristics of examined groups.

Examined Group	Total *N* (100)	Age Groups *n* (%)			Gender *n* (%)		Education *n* (%)		
		≤30	30–40	>40	Women	Men	Primary	Secondary	Higher
MS total (1)	61	14 (22.9)	28 (45.9)	19 (31.2)	42 (68.9)	19 (31.1)	3 (4.9)	25 (41)	33 (54.1)
MS I (1a)	30	12 (40.0)	13 (43.3)	5 (16.7)	21 (70.0)	9 (30.0)	0 (0)	12 (40)	18 (60)
MS II (1b)	31	2 (6.5)	15 (48.4)	14 (45.2)	21 (67.7)	10 (32.3)	3 (9.7)	13 (41.9)	15 (48.4)
Control (2)	21	9 (42.9)	7 (33.3)	5 (23.8)	16 (76.2)	5 (23.8)	3 (14.3)	7 (33.3)	11 (52.4)
1 vs. 2 ^A^		0.23			0.52		0.39		
1a vs. 1b ^B^		<0.01	0.69	<0.05	0.85	0.85	0.08	0.88	0.36
1a vs. 2 ^B^		0.84	0.47	0.53	0.63	0.63	0.05	0.63	0.59
1b vs. 2 ^B^		<0.01	0.28	0.12	0.51	0.51	0.61	0.53	0.78

Data presented as: n (%)—numbers and frequency; ^A^—*p*-value by the χ^2^ test/the Fisher test, ^B^—*p*-value by the post-hoc Bonferroni test.

**Table 2 jcm-12-00093-t002:** Clinical characteristics of patients with multiple sclerosis (MS).

Multiple Sclerosis	MS Total *n* = 61			MS I *n* = 30			MS II *n* = 31			*p*-Value ^A^ MS I vs. MS II
	Min ÷ Max	X ± SD	M (IQR)	Min ÷ Max	X ± SD	M (IQR)	Min ÷ Max	X ± SD	M (IQR)	
Age of first symptoms [years]	16 ÷ 44	28.4 ± 6.5	28 (10)	19 ÷ 44	30.9 ± 6.7	30.5 (11)	16 ÷ 35	26.0 ± 5.3	26 (8)	<0.01
Age of diagnosis	18 ÷ 44	29.8 ± 6.2	29 (10)	19 ÷ 44	31.4 ± 6.6	31.5 (11)	18 ÷ 39	28.3 ± 5.4	28 (10)	<0.05
Disease duration (time from first symptoms) [years]	1 ÷ 28	8.0 ± 6.1	10 (10)	1 ÷ 5	2.8 ± 1.5	2 (3)	10 ÷ 28	13.2 ± 4.0	12 (5)	<0.0001
Number of relapses	1 ÷ 7	3.3 ± 1.5	3 (2)	1 ÷ 5	2.4 ± 1.0	2 (1)	2 ÷ 7	4.2 ± 1.4	4 (2)	<0.0001
EDSS [points]	1 ÷ 4.5	2.5 ± 1.1	2 (1)	1 ÷ 3	1.8 ± 0.6	2 (0.5)	1.5 ÷ 4.5	3.1 ± 1.0	3 (2)	<0.0001
Functional Systems Scores:										
Visual [points]	0 ÷ 2	0.3 ± 0.5	0 (0)	0 ÷ 1	0.2 ± 0.4	0 (0)	0 ÷ 2	0.3 ± 0.6	0 (1)	0.64
Brainstem [points]	0 ÷ 3	1.0 ± 0.9	1 (2)	0 ÷ 2	0.7 ± 0.8	0.5 (1)	0 ÷ 3	1.2 ± 1.0	1 (2)	<0.05
Pyramidal [points]	0 ÷ 4	1.4 ± 1.0	1 (1)	0 ÷ 3	0.9 ± 0.9	1 (2)	0 ÷ 4	1.9 ± 1.0	2 (1)	<0.001
Cerebellar [points]	0 ÷ 3	0.7 ± 0.9	1 (1)	0 ÷ 2	0.4 ± 0.6	0 (1)	0 ÷ 3	1.0 ± 1.0	1 (1)	<0.05
Sensory [points]	0 ÷ 3	1.1 ± 1.0	1 (2)	0 ÷ 2	0.6 ± 0.9	0 (1)	0 ÷ 3	1.6 ± 0.9	1 (1)	<0.001
Bowel and bladder [points]	0 ÷ 3	0.6 ± 0.9	0 (1)	0 ÷ 2	0.2 ± 0.5	0 (0)	0 ÷ 3	1.0 ± 1.0	1 (2)	<0.001
Cerebral/Mental [points]	0 ÷ 1	0.2 ± 0.4	0 (0)	0 ÷ 1	0.1 ± 0.3	0 (0)	0 ÷ 1	0.3 ± 0.4	0 (1)	0.20
Ambulation Scoring [points]	0 ÷ 2	0.8 ± 0.6	1 (1)	0 ÷ 1	0.6 ± 0.5	1 (1)	0 ÷ 2	1.0 ± 0.6	1 (0)	<0.05

Data presented as: min ÷ max—minimum and maximum; X ± SD—mean and standard deviation; M (IQR)—median and interquartile range; ^A^—*p*-value by the *t*-Student test/the U Mann-Whitney test; MS—Multiple sclerosis; MS I—patients with short duration of the disease; MS II—patients with long duration of the disease; EDSS—Expanded Disability Status Scale.

**Table 3 jcm-12-00093-t003:** OCT results in the examined groups.

Examined Group	pRNFL Total	pRNFL Inferior	pRNFL Temporalis	pRNFL Nasal	pRNFL Superior	mRNFL	GCIPL	mRNFL + GCIPL
MS Total (1)	92.2 ± 11.1	111.3 ± 18.1	62.7 ± 13.7	79.0 ± 13.4	109.1 ± 14.0	30.0 ± 7.2	61.3 ± 8.3	91.5 ± 14.5
MS I (1a)	97.3 ± 10.5	118.4 ± 14.7	68.3 ± 12.3	81.7 ± 14.9	112.8 ± 15.1	33.8 ± 5.7	65.5 ± 7.2	99.1 ± 11.5
MS II (1b)	87.2 ± 9.5	104.5 ± 18.7	57.3 ± 13.0	76.4 ± 11.5	105.6 ± 12.2	26.4 ± 6.7	57.2 ± 7.3	84.0 ± 13.5
Control (2)	102.2 ± 7.3	126.4 ± 13.0	75.2 ± 7.5	77.2 ± 13.9	119.9 ± 12.1	36.6 ± 6.3	67.3 ± 6.0	104.3 ± 10.8
1 vs. 2 ^A^	<0.001	<0.01	<0.01	0.68	<0.01	<0.001	<0.01	<0.001
1a vs. 1b ^B^	<0.01	<0.01	<0.01	0.37	0.15	<0.001	<0.001	<0.001
1a vs. 2 ^B^	0.23	0.26	0.24	0.62	0.20	0.35	0.68	0.37
1b vs. 2 ^B^	<0.0001	<0.001	<0.001	0.98	<0.01	<0.0001	<0.0001	<0.0001

Data presented as: X ± SD—mean and standard deviation; ^A^—*p*-values by the *t*-Student test/U Mann-Whitney test; ^B^—*p*-values by post-hoc tests of ANOVA/the Kruskal–Wallis test; MS—Multiple sclerosis; MS I (1a)—patients with short duration of the disease; MS II (1b)—patients with long duration of the disease; pRNFL (thickness of peripapillary retinal nerve fiber layer, assessed in total and in four quadrants: inferior, superior, temporalis and nasal), mRNFL (thickness of macular retinal nerve fiber layer), GCIPL (thickness of macular ganglion cell—GCL and inner plexiform layer—IPL; a parameter that combines GCL and IPL), mRNFL + GCIPL (a parameter that combines the value of macular RNFL and GCIPL).

**Table 4 jcm-12-00093-t004:** Results of psychometric and neurophysiological tests.

Examined Group	Picture Completion (Points)	Digit Symbol (Points)	MoCA Visuospatial (Points)	MoCA Naming (Points)	MoCA Attention (Points)	MoCA Language (Points)	MoCA Abstraction (Points)	MoCA Memory (Points)	MoCA Orientation (Points)	MoCA Total (Points)	P300 Latency (ms)
MS Total (1)	27.4 ± 4.4	52.6 ± 12.7	3.8 ± 1.3	3 ± 0.2	5.3 ± 0.9	2.7 ± 0.6	1.9 ± 0.3	3.9 ± 1.3	6 ± 0	26.8 ± 2.9	318.1 ± 28.3
MS I (1a)	28.9 ± 3.4	61.0 ± 9.3	4.2 ± 1.2	3 ± 0.2	5.6 ± 0.8	2.9 ± 0.3	2.0 ± 0.0	4.1 ± 1.3	6 ± 0	27.7 ± 2.6	312.6 ± 28.6
MS II (1b)	25.9 ± 4.9	44.5 ± 1.0	3.4 ± 1.2	3 ± 0.2	5.1 ± 0.9	2.5 ± 0.7	1.8 ± 0.5	3.8 ± 1.3	6 ± 0	25.9 ± 2.9	323.5 ± 27.5
Control (2)	31.0 ± 3.0	56.4 ± 7.1	4.9 ± 0.3	3 ± 0.0	5.9 ± 0.4	3.0 ± 0.2	2.0 ± 0.0	4.8 ± 0.4	6 ± 0	29.5 ± 0.9	297.3 ± 16.2
1 vs. 2 ^A^	<0.001	0.19	<0.001	0.83	<0.05	0.14	0.66	<0.01	0.99	<0.0001	<0.01
1a vs. 1b ^B^	<0.05	<0.0001	<0.05	0.99	0.1	0.09	0.99	0.98	0.99	<0.05	0.40
1a vs. 2 ^B^	0.10	0.66	0.12	0.99	0.85	0.99	0.99	0.13	0.99	<0.05	0.10
1b vs. 2 ^B^	<0.0001	<0.01	<0.0001	0.99	<0.01	0.05	0.99	<0.05	0.99	<0.0001	<0.01

Data presented as: X ± SD—mean and standard deviation; ^A^—*p*-value by the *t*-Student test/the U Mann-Whitney test; ^B^—*p*-values by post-hoc tests of ANOVA/the Kruskal–Wallis test; MS—Multiple sclerosis; MS I (1a)—patients with short duration of the disease; MS II (1b)—patients with long duration of the disease; MoCA—Montreal Cognitive Assessment test.

**Table 5 jcm-12-00093-t005:** Correlation between the OCT parameters and the results of psychometric tests.

Parameter	Group	P300 Latency	Digit Symbol	Picture Completion	MoCA Visuo-Spatial	MoCA Naming	MoCA Attention	MoCA Language	MoCA Abstraction	MoCA Memory	MoCA Orientation
pRNFL Total	MS Total (1)	0.12	0.21	0.02	−0.07	−0.29 *	0.12	−0.05	−0.18	−0.18	−0.08
	MS I (1a)	0.08	0.01	−0.10	−0.31	−0.25	−0.15	−0.19	0.00	−0.19	−0.20
	MS II (1b)	0.31	−0.28	−0.13	−0.09	−0.32	0.02	−0.21	−0.44 *	−0.37	−0.35
	Control (2)	−0.42	−0.20	0.02	0.17	0.00	0.01	−0.09	0.00	−0.04	0.14
pRNFL Inferior	MS Total (1)	0.07	0.22	0.14	0.08	−0.16	0.09	0.08	−0.15	0.04	0.11
	MS I (1a)	−0.15	0.08	0.16	−0.04	0.09	0.02	0.00	0.00	0.15	0.13
	MS II (1b)	0.30	−0.14	−0.06	−0.05	−0.30	−0.13	−0.02	−0.36	−0.17	−0.19
	Control (2)	−0.32	−0.13	0.08	0.05	0.00	0.13	0.04	0.00	0.05	0.17
pRNFL Temporalis	MS Total (1)	−0.03	0.25	−0.04	0.01	−0.24	0.26	−0.04	−0.18	−0.11	0.00
	MS I (1a)	0.05	0.10	−0.14	−0.05	−0.25	−0.02	0.05	0.00	−0.08	−0.07
	MS II (1b)	0.04	−0.25	−0.29	−0.19	−0.25	0.20	−0.36	−0.44 *	−0.32	−0.35
	Control (2)	−0.31	0.22	−0.02	0.17	0.00	−0.46	0.00	0.00	0.08	−0.15
pRNFL Nasal	MS Total (1)	0.19	0.04	0.06	−0.07	−0.03	0.18	0.03	−0.08	0.00	−0.02
	MS I (1a)	0.24	0.01	−0.21	−0.01	0.20	−0.08	0.23	0.00	0.09	−0.02
	MS II (1b)	0.19	−0.29	0.15	−0.22	−0.23	0.29	−0.19	−0.23	−0.22	−0.23
	Control (2)	−0.12	−0.03	0.39	0.23	0.00	0.34	0.00	0.00	0.46	0.64 *
pRNFL Superior	MS Total (1)	0.18	0.21	−0.05	−0.15	−0.31 *	−0.08	−0.05	−0.16	−0.35 *	−0.22
	MS I (1a)	0.31	0.05	−0.19	−0.32	−0.33	−0.30	−0.19	0.00	−0.48 *	−0.36
	MS II (1b)	0.10	0.03	−0.07	−0.15	−0.33	−0.19	−0.18	−0.30	−0.35	−0.37
	Control (2)	−0.15	−0.45	−0.23	0.02	0.00	−0.28	−0.39	0.00	−0.22	−0.14
mRNFL	MS Total (1)	−0.18	0.33 *	0.10	0.06	−0.28 *	0.21	0.17	−0.02	0.05	0.18
	MS I (1a)I	−0.01	−0.04	−0.06	−0.14	−0.21	−0.01	−0.02	0.00	−0.01	0.02
	MS II (1b)	−0.27	0.00	−0.19	−0.15	−0.33	−0.05	0.04	−0.26	−0.06	−0.07
	Control (2)	−0.12	0.02	0.08	0.29	0.00	−0.08	0.13	0.00	−0.11	−0.03
GCIPL	MS Total (1)	−0.09	0.27	−0.01	0.03	−0.15	0.12	−0.02	−0.10	0.04	0.10
	MS I (1a)	0.04	−0.11	−0.13	−0.15	0.07	0.01	−0.14	0.00	0.23	0.06
	MS II (1b)	−0.12	−0.10	−0.22	−0.22	−0.30	−0.17	−0.23	−0.37	−0.23	−0.28
	Control (2)	−0.31	−0.22	0.19	0.20	0.00	0.00	−0.06	0.00	−0.02	0.13
GCIPL +mRNFL	MS Total (1)	−0.10	0.26	0.01	0.01	−0.23	0.15	0.01	−0.16	−0.01	0.09
	MS I (1a)I	0.06	−0.19	−0.13	−0.16	−0.09	−0.08	−0.17	0.00	0.06	−0.01
	MS II (1b)	−0.21	−0.10	−0.23	−0.22	−0.33	−0.09	−0.14	−0.42 *	−0.18	−0.21
	Control (2)	−0.17	−0.18	0.13	0.19	0.00	−0.36	−0.19	0.00	−0.13	−0.19

Values of the Spearman correlation coefficient R and the significance test; *—*p* < 0.05; MS—Multiple sclerosis; MS I (1a)—patients with short duration of the disease; MS II (1b)—patients with long duration of the disease; MoCA—Montreal Cognitive Assessment test; pRNFL (thickness of peripapillary retinal nerve fiber layer, assessed in total and in 4 quadrants: inferior, superior, temporalis and nasal), mRNFL (thickness of macular retinal nerve fiber layer), GCIPL (thickness of macular ganglion cell—GCL and inner plexiform layer—IPL; a parameter that combines GCL and IPL), mRNFL + GCIPL (a parameter that combines the value of macular RNFL and GCIPL).

**Table 6 jcm-12-00093-t006:** Differences in OCT parameters due to gender, age, and education level.

Parameter	Group	Gender			Age Groups				Education Level			
		Women	Men	*p*-Value ^A^	≤30	30–40	>40	*p*-Value ^B^	Primary	Secondary	Higher	*p*-Value ^B^
pRNFL Total	MS Total (1)	92 ± 11	93 ± 11	0.52	98 ± 9	94 ± 10	85 ± 11	<0.01	93 ± 13	94 ± 9	90 ± 12	0.46
	MS I (1a)	98 ± 12	97 ± 6	0.87	98 ± 9	99 ± 9	91 ± 17	0.46	-	97 ± 9	98 ± 12	-
	MS II (1b)	86 ± 7	89 ± 14	0.79	100 ± 12	89 ± 8	83 ± 9	<0.05	93 ± 13	92 ± 9	83 ± 8	<0.05
	Control (2)	101 ± 8	106 ± 4	0.34	102 ± 7	106 ± 4	97 ± 10	0.13	105 ± 0	104 ± 10	101 ± 6	0.58
	1 vs. 2 ^A^/^C^	<0.01	<0.05	<0.01	0.38	<0.0001	0.08	<0.0001	0.31	<0.05	<0.01	<0.05
	1a vs. 1b vs. 2 ^B^/^C^	<0.001	0.05	<0.0001	0.67	<0.001	0.10	<0.0001	-	<0.05	<0.0001	-
pRNFL Inferior	MS Total (1)	110 ± 19	113 ± 17	0.68	121 ± 14	115 ± 15	98 ± 19	<0.01	112 ± 20	115 ± 13	108 ± 21	0.41
	MS I (1a)	119 ± 17	117 ± 10	0.71	119 ± 13	123 ± 13	105 ± 19	0.11	-	118 ± 10	119 ± 18	-
	MS II (1b)	102 ± 17	109 ± 22	0.88	133 ± 19	108 ± 13	96 ± 20	<0.05	112 ± 20	112 ± 15	98 ± 19	0.13
	Control (2)	126 ± 15	127 ± 2	0.96	127 ± 13	130 ± 13	120 ± 15	0.50	136 ± 13	127 ± 15	124 ± 12	0.51
	1 vs. 2 ^A^/^C^	<0.01	<0.05	<0.05	0.33	<0.05	0.06	<0.0001	0.24	0.06	<0.05	<0.05
	1a vs. 1b vs. 2 ^B^/^C^	<0.001	0.09	<0.01	0.26	<0.01	0.12	<0.0001	-	0.1	<0.001	-
pRNFL Temporalis	MS Total (1)	63 ± 15	62 ± 11	0.85	72 ± 12	63 ± 12	56 ± 14	<0.01	62 ± 17	64 ± 10	62 ± 16	0.95
	MS I (1a)	70 ± 15	65 ± 5	0.22	72 ± 12	65 ± 9	66 ± 20	0.48	-	66 ± 10	70 ± 14	-
	MS II (1b)	57 ± 13	58 ± 14	0.84	70 ± 13	60 ± 14	52 ± 10	0.11	62 ± 17	60 ± 9	54 ± 15	0.43
	Control (2)	75 ± 8	76 ± 7	0.74	76 ± 6	77 ± 9	69 ± 6	0.42	80 ± 0	71 ± 5	77 ± 8	0.32
	1 vs. 2 ^A^/^C^	<0.05	<0.05	<0.05	0.39	<0.05	0.22	<0.001	0.45	0.18	<0.05	0.08
	1a vs. 1b vs. 2 ^B^/^C^	<0.01	<0.05	<0.001	0.68	<0.05	0.09	<0.001	-	0.14	<0.001	-
pRNFL Nasal	MS Total (1)	79 ± 15	79 ± 11	0.99	79 ± 12	81 ± 13	76 ± 16	0.56	80 ± 11	81 ± 12	77 ± 15	0.64
	MS I (1a)	82 ± 18	81 ± 8	0.79	79 ± 13	86 ± 13	78 ± 23	0.46	-	81 ± 13	82 ± 17	-
	MS II (1b)	76 ± 11	77 ± 14	1	81 ± 4	77 ± 11	76 ± 14	0.84	79 ± 11	82 ± 11	72 ± 11	0.13
	Control (2)	75 ± 14	86 ± 12	0.35	77 ± 9	84 ± 15	61 ± 12	0.12	72 ± 0	78 ± 22	77 ± 11	0.93
	1 vs. 2 ^A^/^C^	0.41	0.37	0.64	0.74	0.63	0.20	0.34	0.62	0.70	0.99	0.94
	1a vs. 1b vs. 2 ^B^/^C^	0.4	0.53	0.60	0.92	0.17	0.44	0.36	-	0.92	0.15	-
mRNFL	MS Total (1)	31 ± 7	29 ± 8	0.90	33 ± 7	31 ± 5	27 ± 9	<0.05	28 ± 10	30 ± 5	31 ± 9	0.82
	MS I (1a)	34 ± 7	33 ± 3	0.37	34 ± 7	34 ± 4	32 ± 7	0.75	-	33 ± 4	35 ± 7	-
	MS II (1b)	27 ± 4	25 ± 10	0.89	27 ± 6	28 ± 5	25 ± 9	0.6	28 ± 10	27 ± 4	26 ± 8	0.90
	Control (2)	37 ± 7	35 ± 1	0.73	37 ± 5	36 ± 6	36 ± 10	0.97	35 ± 1	39 ± 10	36 ± 3	0.55
	1 vs. 2 ^A^/^C^	<0.01	0.07	<0.01	0.20	<0.05	0.07	<0.01	0.43	0.05	<0.01	<0.05
	1a vs. 1b vs. 2 ^B^/^C^	<0.01	<0.05	<0.0001	0.17	<0.001	0.08	<0.001	-	<0.01	<0.001	-
GCIPL	MS Total (1)	61 ± 8	62 ± 9	0.45	64 ± 9	63 ± 8	57 ± 7	<0.05	59 ± 10	62 ± 7	61 ± 9	0.72
	MS I (1a)	66 ± 8	65 ± 6	0.85	65 ± 8	68 ± 6	60 ± 8	0.17	-	66 ± 5	66 ± 9	-
	MS II (1b)	57 ± 5	59 ± 11	0.49	61 ± 18	58 ± 6	56 ± 7	0.53	59 ± 10	59 ± 8	56 ± 7	0.53
	Control (2)	67 ± 7	69 ± 1	0.62	67 ± 7	69 ± 4	64 ± 8	0.36	69 ± 1	69 ± 8	66 ± 5	0.62
	1 vs. 2 ^A^/^C^	<0.05	0.17	<0.05	0.41	<0.05	0.09	<0.01	0.28	0.06	0.09	0.11
	1a vs. 1b vs. 2 ^B^/^C^	<0.001	0.16	<0.001	0.55	<0.001	0.15	<0.001	-	<0.05	<0.001	-
GCIPL +mRNFL	MS Total (1)	92 ± 14	92 ± 17	0.46	97 ± 14	93 ± 12	83 ± 16	<0.05	91 ± 25	92 ± 11	91 ± 16	0.94
	MS I (1a)	100 ± 14	98 ± 6	0.61	99 ± 13	102 ± 8	92 ± 14	0.28	-	99 ± 7	100 ± 14	-
	MS II (1b)	84 ± 8	84 ± 22	0.60	89 ± 24	86 ± 10	81 ± 16	0.52	91 ± 25	86 ± 12	82 ± 14	0.62
	Control (2)	104 ± 12	105 ± 1	0.42	105 ± 10	106 ± 8	100 ± 18	0.68	105 ± 1	108 ± 17	102 ± 6	0.54
	1 vs. 2 ^A^	<0.01	0.11	<0.05	0.21	<0.05	0.08	<0.01	0.57	<0.05	<0.01	<0.05
	1a vs. 1b vs. 2 ^B^/^C^	<0.001	0.12	<0.0001	0.29	<0.0001	0.12	<0.0001	-	<0.01	<0.001	-

Data presented as: X ± SD—mean and standard deviation; ^A^—*p*-values by the *t*-Student test/U Mann-Whitney test; ^B^—*p*-values by the tests of ANOVA/the Kruskal–Wallis test; ^C^—*p*-values by parametric/non-parametric two-way analysis of variance; MS—Multiple sclerosis; MS I (1a)- patients with short duration of the disease; MS II (1b)—patients with long duration of the disease; pRNFL (thickness of peripapillary retinal nerve fiber layer, assessed in total and in four quadrants: inferior, superior, temporalis and nasal), mRNFL (thickness of macular retinal nerve fiber layer), GCIPL (thickness of macular ganglion cell—GCL and inner plexiform layer—IPL; a parameter that combines GCL and IPL), mRNFL + GCIPL (a parameter that combines the value of macular RNFL and GCIPL).

**Table 7 jcm-12-00093-t007:** Differences in OCT parameters due to groups by MS duration and MS treatment.

Parameter	Group	First MS Symptoms			MS Treatment					
		≤30	>30	*p*-Value ^A^	Interferon Beta	Natalizumab	Fingolimod	Glatiramer Acetate	No Treatment	*p*-Value ^B^
pRNFL Total	MS Total (1)	93 ± 2	91 ± 12	0.53	93 ± 12	87 ± 12	92 ± 1	99 ± 2	10 ± 2	0.45
	MS I (1a)	100 ± 9	94 ± 12	0.19	96 ± 1	113 ± 0	97 ± 9	106 ± 2	94 ± 15	0.40
	MS II (1b)	88 ± 9	82 ± 9	0.20	90 ± 12	84 ± 6	86 ± 9	87 ± 0	88 ± 6	0.83
	1a vs. 1b ^A^/^C^	<0.01	0.05	-	0.16	-	0.20	-	0.83	-
pRNFL Inferior	MS Total (1)	113 ± 17	109 ± 22	0.45	114 ± 17	102 ± 26	109 ± 14	128 ± 22	108 ± 12	0.47
	MS I (1a)	122 ± 14	115 ± 15	0.25	117 ± 15	133 ± 0	112 ± 12	140 ± 15	117 ± 7	0.24
	MS II (1b)	107 ± 16	94 ± 29	0.15	109 ± 19	98 ± 24	106 ± 16	106 ± 0	98 ± 8	0.80
	1a vs. 1b ^A^/^C^	<0.01	0.06	-	0.19	-	0.77	-	0.08	-
pRNFL Temporalis	MS Total (1)	64 ± 14	61 ± 14	0.52	64 ± 13	54 ± 17	62 ± 10	78 ± 4	62 ± 15	0.11
	MS I (1a)	71 ± 11	65 ± 13	0.18	68 ± 13	90 ± 0	66 ± 11	77 ± 2	60 ± 9	0.23
	MS II (1b)	59 ± 13	52 ± 12	0.28	58 ± 12	49 ± 10	58 ± 7	82 ± 0	64 ± 22	0.26
	1a vs. 1b ^A^/^C^	<0.01	0.07	-	<0.05	-	0.15	-	0.66	-
pRNFL Nasal	MS Total (1)	78 ± 13	80 ± 15	0.61	79 ± 15	80 ± 11	81 ± 14	76 ± 10	78 ± 2	0.98
	MS I (1a)	81 ± 14	83 ± 16	0.70	80 ± 17	98 ± 0	91 ± 7	82 ± 0	73 ± 8	0.27
	MS II (1b)	77 ± 11	75 ± 14	0.80	77 ± 13	78 ± 10	71 ± 11	65 ± 0	82 ± 14	0.59
	1a vs. 1b ^A^/^C^	0.33	0.39	-	0.52	-	<0.05	-	0.38	-
pRNFL Superior	MS Total (1)	111 ± 13	105 ± 15	0.14	110 ± 14	108 ± 11	109 ± 2	111 ± 1	105 ± 23	0.94
	MS I (1a)	117 ± 13	108 ± 16	0.12	111 ± 15	129 ± 0	115 ± 1	117 ± 5	112 ± 30	0.59
	MS II (1b)	107 ± 12	98 ± 13	0.15	109 ± 15	105 ± 8	103 ± 8	100 ± 0	97 ± 16	0.59
	1a vs. 1b ^A^/^C^	<0.05	0.27	-	0.98	-	0.11	-	0.38	-
mRNFL	MS Total (1)	29 ± 8	32 ± 5	0.11	31 ± 7	29 ± 7	30 ± 6	32 ± 4	27 ± 11	0.84
	MS I (1a)	34 ± 6	33 ± 5	0.65	34 ± 6	40 ± 0	32 ± 7	33 ± 4	35 ± 1	0.75
	MS II (1b)	26 ± 7	3 ± 5	0.17	27 ± 7	27 ± 5	27 ± 4	29 ± 0	19 ± 11	0.67
	1a vs. 1b ^A^/^C^	<0.001	0.27	-	<0.05	-	0.25	-	0.08	-
GCIPL	MS Total (1)	61 ± 9	62 ± 7	0.50	63 ± 8	57 ± 7	60 ± 8	63 ± 7	59 ± 11	0.40
	MS I (1a)	67 ± 8	64 ± 7	0.38	66 ± 8	69 ± 0	62 ± 7	66 ± 6	68 ± 5	0.87
	MS II (1b)	57 ± 8	58 ± 3	0.70	60 ± 8	55 ± 5	57 ± 8	57 ± 0	51 ± 10	0.69
	1a vs. 1b ^A^/^C^	<0.001	0.10	-	0.07	-	0.31	-	0.08	-
GCIPL +mRNFL	MS Total (1)	90 ± 16	96 ± 10	0.10	95 ± 14	86 ± 13	89 ± 13	95 ± 10	87 ± 22	0.49
	MS I (1a)	101 ± 12	97 ± 2	0.48	99 ± 12	108 ± 0	95 ± 15	99 ± 2	103 ± 6	0.75
	MS II (1b)	83 ± 14	91 ± 5	0.27	88 ± 14	82 ± 9	84 ± 11	88 ± 0	71 ± 3	0.57
	1a vs. 1b ^A^/^C^	<0.001	0.28	-	0.07	-	0.31	-	0.08	-

Data presented as: X ± SD—mean and standard deviation; **^A^**—*p*-values by the *t*-Student test/U Mann-Whitney test; ^B^—*p*-values the tests of ANOVA/the Kruskal–Wallis test; ^C^—*p*-values by parametric/non-parametric two-way analysis of variance; MS—Multiple sclerosis; MS I (1a)- patients with short duration of the disease; MS II (1b)—patients with long duration of the disease; pRNFL (thickness of peripapillary retinal nerve fiber layer, assessed in total and in four quadrants: inferior, superior, temporalis and nasal), mRNFL (thickness of macular retinal nerve fiber layer), GCIPL (thickness of macular ganglion cell—GCL and inner plexiform layer—IPL; a parameter that combines GCL and IPL), mRNFL + GCIPL (a parameter that combines the value of macular RNFL and GCIPL).

**Table 8 jcm-12-00093-t008:** Correlation between the OCT parameters and EDSS score, number of relapses, MS duration (time from first symptoms).

Parameter	Group	EDSS Score	Number of Relapses	MS Duration	Model ^A^ *p*-Value B Coefficient; *p*-Value
pRNFL Total	MS Total (1)	−0.36 *	−0.25	−0.37 *	<0.001 Duration: −0.80; <0.001
	MS I (1a)	−0.03	0.05	0.29	
	MS II (1b)	−0.16	0.03	−0.16	
pRNFL Inferior	MS Total (1)	−0.30 *	−0.16	−0.40 *	<0.001 Age: −1.20; <0.001
	MS I (1a)	−0.13	0.05	0.20	
	MS II (1b)	−0.01	0.18	−0.27	
pRNFL Temporalis	MS Total (1)	−0.41 *	−0.27	−0.36 *	<0.001 Age: −0.78; <0.01 EDSS: −0.80, <0.001
	MS I (1a)	−0.04	0.03	0.16	
	MS II (1b)	−0.24	−0.09	−0.18	
pRNFL Nasal	MS Total (1)	−0.02	0.01	−0.08	0.25 Duration: −0.34; 0.25
	MS I (1a)	0.37	0.25	0.41 *	
	MS II (1b)	−0.10	0.07	0.03	
pRNFL Superior	MS Total (1)	−0.26	−0.16	−0.25	0.03 Age: −0.53; 0.06 EDSS: −2.86; 0.12
	MS I (1a)	−0.07	0.07	0.11	
	MS II (1b)	−0.22	−0.04	−0.18	
mRNFL	MS Total (1)	−0.56 *	−0.24	−0.52 *	<0.0001 Duration: −0.67; <0.0001
	MS I (1a)	−0.13	0.23	0.26	
	MS II (1b)	−0.41 *	0.04	−0.50 *	
GCIPL	MS Total (1)	−0.44 *	−0.34 *	−0.52 *	<0.0001 Duration: −0.76; <0.0001
	MS I (1a)	0.06	−0.01	0.13	
	MS II (1b)	−0.38 *	−0.13	−0.51 *	
GCIPL + mRNFL	MS Total (1)	−0.51 *	−0.32 *	−0.54 *	<0.0001 Duration: −1.40; <0.0001
	MS I (1a)	−0.04	0.12	0.19	
	MS II (1b)	−0.44 *	−0.08	−0.52 *	

Values of the Spearman correlation coefficient R and the significance test; *—*p* < 0.05; ^A^—multiple backward stepwise regression model results: p-value of models significance test, B- regression coefficients and the significance test; MS—Multiple sclerosis; MS I (1a)—patients with short duration of the disease; MS II (1b)—patients with long duration of the disease; pRNFL (thickness of peripapillary retinal nerve fiber layer, assessed in total and in four quadrants: inferior, superior, temporalis and nasal), mRNFL (thickness of macular retinal nerve fiber layer), GCIPL (thickness of macular ganglion cell—GCL and inner plexiform layer—IPL; a parameter that combines GCL and IPL), mRNFL + GCIPL (a parameter that combines the value of macular RNFL and GCIPL).

**Table 9 jcm-12-00093-t009:** Correlation between the OCT parameters and functional systems scores.

Parameter	Group	Visual Functions	Brainstem Functions	Pyramidal System Functions	Cerebellar Functions	Sensory Functions	Bladder and Bowel Functions	Cognitive Functions	Ambulation Index
pRNFL Total	MS Total (1)	−0.01	−0.38 *	−0.17	−0.29 *	−0.25	−0.45 *	−0.19	−0.10
	MS I (1a)	−0.13	−0.22	0.25	−0.11	0.09	0.11	0.25	0.16
	MS II (1b)	0.07	−0.38 *	−0.05	−0.19	−0.11	−0.45 *	−0.21	−0.05
pRNFL Inferior	MS Total (1)	−0.06	−0.36 *	−0.23	−0.11	−0.18	−0.50 *	−0.19	−0.02
	MS I (1a)	−0.21	−0.23	0.03	0.12	0.17	−0.25	−0.09	0.03
	MS II (1b)	0.12	−0.36	−0.04	−0.02	−0.05	−0.38	−0.11	0.16
pRNFL Temporalis	MS Total (1)	−0.10	−0.32 *	−0.17	−0.30 *	−0.33 *	−0.25	−0.24	−0.24
	MS I (1a)	−0.23	−0.16	0.20	0.08	−0.08	0.39 *	0.25	0.13
	MS II (1b)	−0.03	−0.23	−0.04	−0.36	−0.25	−0.16	−0.24	−0.27
pRNFL Nasal	MS Total (1)	0.03	−0.14	0.06	−0.09	−0.04	−0.33 *	−0.21	0.15
	MS I (1a)	0.10	0.22	0.36	−0.02	0.16	−0.31	−0.20	0.45 *
	MS II (1b)	0.02	−0.38 *	0.07	0.03	−0.04	−0.30	−0.21	0.00
pRNFL Superior	MS Total (1)	0.07	−0.16	−0.19	−0.33 *	−0.15	−0.31 *	−0.13	−0.19
	MS I (1a)	0.02	−0.18	0.13	−0.21	−0.02	0.24	0.33	0.04
	MS II (1b)	0.07	0.08	−0.27	−0.36	0.08	−0.40 *	−0.18	−0.30
mRNFL	MS Total (1)	−0.04	−0.31 *	−0.40 *	−0.34 *	−0.34 *	−0.39 *	−0.15	−0.20
	MS I (1a)	−0.01	−0.06	0.05	−0.14	−0.12	0.18	0.21	−0.03
	MS II (1b)	−0.10	−0.29	−0.38 *	−0.38 *	−0.02	−0.27	−0.04	0.01
GCIPL	MS Total (1)	−0.12	−0.22	−0.28 *	−0.39 *	−0.26	−0.37 *	−0.21	−0.12
	MS I (1a)	−0.28	−0.04	0.23	−0.01	0.04	0.25	−0.07	0.24
	MS II (1b)	−0.02	−0.04	−0.29	−0.51 *	0.02	−0.22	−0.05	−0.16
GCIPL + mRNFL	MS Total (1)	−0.05	−0.26	−0.35 *	−0.38 *	−0.33 *	−0.39 *	−0.22	−0.18
	MS I (1a)	−0.21	−0.06	0.12	−0.07	−0.03	0.18	0.09	0.10
	MS II (1b)	−0.03	−0.16	−0.36	−0.49 *	−0.04	−0.24	−0.10	−0.12

Values of the Spearman correlation coefficient R and the significance test; *—*p* < 0.05; MS—Multiple sclerosis; MS I (1a)—patients with short duration of the disease; MS II (1b)—patients with long duration of the disease; pRNFL (thickness of peripapillary retinal nerve fiber layer, assessed in total and in four quadrants: inferior, superior, temporalis and nasal), mRNFL (thickness of macular retinal nerve fiber layer), GCIPL (thickness of macular ganglion cell—GCL and inner plexiform layer—IPL; a parameter that combines GCL and IPL), mRNFL + GCIPL (a parameter that combines the value of macular RNFL and GCIPL).

## Data Availability

The data presented in this study is available on request from the corresponding author. The data is not publicly available due to privacy reasons.

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
