# Peer review of "The Usefulness of Optical Coherence Tomography in Disease Progression Monitoring in Younger Patients with Relapsing-Remitting Multiple Sclerosis: A Single-Centre Study"

_jcm, 2022, doi:10.3390/jcm12010093_

Round 1
Reviewer 1 Report
The manuscript of Torbus and co-authors is scientifically well conducted and well written but should be improved from the point of view of novelty. In the last two years, the inner nuclear layer (INL) has been suggested as a measure of neurodegeneration, and this started a discussion within this field.
See these papers:
1) Sotirchos ES, Gonzalez Caldito N, Filippatou A, Fitzgerald KC, Murphy OC, Lambe J, Nguyen J, Button J, Ogbuokiri E, Crainiceanu CM, Prince JL, Calabresi PA, Saidha S; International Multiple Sclerosis Visual System (IMSVISUAL) Consortium. Progressive Multiple Sclerosis Is Associated with Faster and Specific Retinal Layer Atrophy. Ann Neurol. 2020 Jun;87(6):885-896. doi: 10.1002/ana.25738. Epub 2020 Apr 28. PMID: 32285484; PMCID: PMC8682917.
2) Cordano C, Yiu HH, Oertel FC; University of California, San Francisco MS-EPIC Team, Gelfand JM, Hauser SL, Cree BAC, Green AJ. Retinal INL Thickness in Multiple Sclerosis: A Mere Marker of Neurodegeneration? Ann Neurol. 2021 Jan;89(1):192-193. doi: 10.1002/ana.25933. Epub 2020 Nov 11. PMID: 33067847; PMCID: PMC7990472.
3) Bijvank JA, Uitdehaag BMJ, Petzold A. Interpretation of Longitudinal Changes of the Inner Nuclear Layer in MS. Ann Neurol. 2022 Jul;92(1):154-155. doi: 10.1002/ana.26365. Epub 2022 Apr 28. PMID: 35403739.
4) Cordano C, Yiu HH, Abdelhak A, Beaudry-Richard A, Oertel FC, Green AJ. Reply to "Interpretation of Longitudinal Changes of the Inner Nuclear Layer in MS". Ann Neurol. 2022 Jul;92(1):156. doi: 10.1002/ana.26367. Epub 2022 Apr 21. PMID: 35403744.
I suggest adding INL thickness to the equation and seeing if this measure can help differentiate early MS patients from longer disease duration.
Author Response
First, we would like to thank for this very detailed analysis of the manuscript. We agree with the majority of the Reviewers’ comments. Our replies are written in a red. We made several changes in our manuscript believing they address all the Reviewers’ comments (also marked in red).
Reviewer 1
The manuscript of Torbus and co-authors is scientifically well conducted and well written but should be improved from the point of view of novelty. In the last two years, the inner nuclear layer (INL) has been suggested as a measure of neurodegeneration, and this started a discussion within this field.
See these papers:
1) Sotirchos ES, Gonzalez Caldito N, Filippatou A, Fitzgerald KC, Murphy OC, Lambe J, Nguyen J, Button J, Ogbuokiri E, Crainiceanu CM, Prince JL, Calabresi PA, Saidha S; International Multiple Sclerosis Visual System (IMSVISUAL) Consortium. Progressive Multiple Sclerosis Is Associated with Faster and Specific Retinal Layer Atrophy. Ann Neurol. 2020 Jun;87(6):885-896. doi: 10.1002/ana.25738. Epub 2020 Apr 28. PMID: 32285484; PMCID: PMC8682917.
2) Cordano C, Yiu HH, Oertel FC; University of California, San Francisco MS-EPIC Team, Gelfand JM, Hauser SL, Cree BAC, Green AJ. Retinal INL Thickness in Multiple Sclerosis: A Mere Marker of Neurodegeneration? Ann Neurol. 2021 Jan;89(1):192-193. doi: 10.1002/ana.25933. Epub 2020 Nov 11. PMID: 33067847; PMCID: PMC7990472.
3) Bijvank JA, Uitdehaag BMJ, Petzold A. Interpretation of Longitudinal Changes of the Inner Nuclear Layer in MS. Ann Neurol. 2022 Jul;92(1):154-155. doi: 10.1002/ana.26365. Epub 2022 Apr 28. PMID: 35403739.
4) Cordano C, Yiu HH, Abdelhak A, Beaudry-Richard A, Oertel FC, Green AJ. Reply to "Interpretation of Longitudinal Changes of the Inner Nuclear Layer in MS". Ann Neurol. 2022 Jul;92(1):156. doi: 10.1002/ana.26367. Epub 2022 Apr 21. PMID: 35403744.
I suggest adding INL thickness to the equation and seeing if this measure can help differentiate early MS patients from longer disease duration.
Re.:
Unfortunately, we did not evaluate the inner nuclear layer (INL), and it really would be interesting to see if there is a difference between patients with shorter and longer disease duration. Thank you for your comment, we hope to continue our research in the future.
We add in Limitation section:
Thirdly, we did not evaluate the inner nuclear layer (INL), the parameter that recently has been discussed as a marker of neurodegeneration in MS [42-45], so further study is needed in this field.
We quoted the suggested references.

Reviewer 2 Report
The research presents an interesting analysis of OCT parameters combined with the results of psychometric tests. An unquestionable advantage is the multitude of psychometric tests used in addition to the OCT examination itself. However, there are some issues that require major improvement.
1. The study was conducted on a relatively small number of RRMS patients. Therefore, it should be noted in the title of the manuscript that the invetigated population was limited to this group of patients.
2. The authors in the title, abstract and conclusions emphasize that OCT may be a reliable method of assessing neurdegeneration in the very early stages of the disease. Meanwhile, they did not show significant differences in OCT parameters between the group with the shortest duration of the disease (1a) and the controls. Most of the significant differences in OCT parameters concerned the entire MS group or subgroup 1b when compared to the controls. What do the authors understand in the term ,, very early stage of the disease” and what time limit do they set for its determination?
3. In the introduction and/or discussion section, there is no mention as well as no references to the differences in OCT parameters depending on the MS variant, the type of immunomodulatory treatment applied and the importance of selecting eyes for analysis (with or without a history of optic neuritis) (see: doi: 10.1212/WNL.0000000000003582; doi:10.3390/jcm10132892; doi: 10.1016/S1474-4422(16)00068-5).
4. In the method section, it was not reported how the authors referred to patients who experienced unilateral optic neuritis - which eye they chose for analysis. It is generally assumed that the analysis of OCT parameters in eyes without the history of optic neuritis is more reliable to illustrate the processes of neurodegeneration compared to eyes with the history of optic neuritis. For example, in a study by Martinez-Lapiscina et al. OCT measures were calculated as the mean values of both eyes without optic neuritis for patients without a history of optic neuritis or the values of the non-optic neuritis eyes for patients with previous unilateral optic neuritis. (see doi: 10.1016/S1474-4422(16)00068-5). These issues should be clarified in the manuscript.
7. The results section lacks data on how many patients used disease-modifying therapy as well as how many subjects experienced optic neuritis in the study group (including also 1a and 1b subgroups). Were there any patients who experienced bilateral optic neuritis?
8. The discussion does not refer to the lack of differences in OCT parameters in the early stage of the disease (subgroup 1a) when compared to the controls.
9. There is no mention in the discussion section about the authors' findings that there were no significant differences in OCT parameters between patients treated with various disease-modifying drugs.
10. I do not entirely agree with the conclusion that the educational level influenced the results of OCT. Did the identified differences not result from the older age of people with primary education compared to those with higher education? In my opinion, the authors' conclusions regarding the dependence of OCT parameters on education are too far-reaching.
11. I am surprised the authors did not mention the limitations of the study.
12. Tables contain too much data, which makes them difficult to read. In table 8 the last parameter should be signed as GCIP + mRNFL instead of GCIPL +pRNFL.
Author Response
First, we would like to thank for this very detailed analysis of the manuscript. We agree with the majority of the Reviewers’ comments. Our replies are written in a red. We made several changes in our manuscript believing they address all the Reviewers’ comments (also marked in red).
Reviewer 2
The research presents an interesting analysis of OCT parameters combined with the results of psychometric tests. An unquestionable advantage is the multitude of psychometric tests used in addition to the OCT examination itself. However, there are some issues that require major improvement.
- The study was conducted on a relatively small number of RRMS patients. Therefore, it should be noted in the title of the manuscript that the investigated population was limited to this group of patients.
Re.: The title of the manuscript was changed:
“The usefulness of optical coherence tomography in detection of neurodegeneration in younger patients with relapsing-remitting multiple sclerosis: a single-centre study”
- The authors in the title, abstract and conclusions emphasize that OCT may be a reliable method of assessing neurodegeneration in the very early stages of the disease. Meanwhile, they did not show significant differences in OCT parameters between the group with the shortest duration of the disease (1a) and the controls. Most of the significant differences in OCT parameters concerned the entire MS group or subgroup 1b when compared to the controls. What do the authors understand in the term ,, very early stage of the disease” and what time limit do they set for its determination?
Re.: Thank you for your valuable comment. We removed the expression “very early stage of the disease” from the paper. We changed the title, abstract and conclusions, emphasizing that we studied younger MS patients with different disease durations
- In the introduction and/or discussion section, there is no mention as well as no references to the differences in OCT parameters depending on the MS variant, the type of immunomodulatory treatment applied and the importance of selecting eyes for analysis (with or without a history of optic neuritis) (see: doi: 10.1212/WNL.0000000000003582; doi:10.3390/jcm10132892; doi: 10.1016/S1474-4422(16)00068-5).
Re.:
We mentioned in the Introduction section about differences in RNFL depending on the MS variant:
Researchers observed thinning of RNFL in all quadrants or only in some quadrants (most often temporal, more rarely lower and upper), more severe in progressive forms of MS (secondary progressive, SPMS; and primary progressive, PPMS) when compared to relapsing-remitting MS (RRMS) and clinically isolated syndrome (CIS) [11-15].
In the Discussion section we discussed immunomodulatory treatment vs OCT:
In our study, the use of DMDs had no significant effect on the results of OCT. However, subgroups of patients treated with each drug were small and it could have influenced this observation. Current DMTs are effective in inhibiting neuroinflammation and reducing the rate of clinical relapses but their neuroprotective effects are less well defined. Data on the relationship between OCT results and DMDs are unclear. Knier et al. analyzed the association between OCT changes and DMT treatment in patients with MS [37]. The authors did not find any difference between patients with or without DMDs but they examined only subjects with first line drugs (interferon beta, glatiramer acetate, dimethyl fumarate, and teriflunomide). On the other hand, You at al. observed that progressive loss of retinal ganglion cells was more pronounced in patients treated with interferon beta than other DMTs (glatiramer acetate, fingolimod, dimethyl fumarate, natalizumab, alemtuzumab, rituximab, and ocrelizumab) [38]. Zivadinov et al. noticed that glatiramer acetate might have a neuroprotective effect against RNFL loss [39]. Button et al. found that GCIPL thinning in patients on both in-terferon beta and glatiramer acetate is faster than in patients on natalizumab [40].
In the limitation section we mention about the importance of selecting eyes for analysis (with or without a history of optic neuritis).
Secondly, although some researchers believe [37] that the analysis of OCT parameters in eyes without the history of ON is more reliable to illustrate the processes of neuro-degeneration than in eyes with the history of ON…
We have also quoted the suggested references.
- In the method section, it was not reported how the authors referred to patients who experienced unilateral optic neuritis - which eye they chose for analysis. It is generally assumed that the analysis of OCT parameters in eyes without the history of optic neuritis is more reliable to illustrate the processes of neurodegeneration compared to eyes with the history of optic neuritis. For example, in a study by Martinez-Lapiscina et al. OCT measures were calculated as the mean values of both eyes without optic neuritis for patients without a history of optic neuritis or the values of the non-optic neuritis eyes for patients with previous unilateral optic neuritis. (see doi: 10.1016/S1474-4422(16)00068-5). These issues should be clarified in the manuscript.
Re.:
It is described in the Material and methods section that both of the patient's eyes were examined. However, patients with optic neuritis (ON) in the last 6 months or other relapse in the last month prior to the recruitment were excluded from the study. We also mentioned in the Discussion section: We examined a total of 164 eyes - 122 eyes of MS patients and 42 eyes in the control group.
We add also in the Limitation section:
Secondly, although some researchers believe [31,41] that the analysis of OCT parameters in eyes without the history of ON is more reliable to illustrate the processes of neurodegeneration than in eyes with the history of ON, we assessed both eyes of each patient regardless of the history of ON. However, patients with ON in the last 6 months prior to the recruitment were excluded from the study.
- The results section lacks data on how many patients used disease-modifying therapy as well as how many subjects experienced optic neuritis in the study group (including also 1a and 1b subgroups). Were there any patients who experienced bilateral optic neuritis?
Re.:
We describe DMD used by patients in the Results section. We also added how many patients had a history of optic neuritis:
The patients were treated with interferon beta (n=28; 16 in 1a and 12 in 1b), glatiramer acetate (n=3; 2 in 1a and 1 in 1b), fingolimod (n=8; 4 in 1a and 4 in 1b) and natali-zumab (n=8; 1 in 1a and 7 in 1b). However, 14 patients (7 in 1a and 7 in 1b) did not re-ceive disease modifying drugs (DMD). 35 patients (15 in 1a and 20 in 1b) had a history of unilateral optic neuritis (nobody experienced bilateral optic neuritis).
- The discussion does not refer to the lack of differences in OCT parameters in the early stage of the disease (subgroup 1a) when compared to the controls.
Re.:
We mentioned in the Discussion:
Interestingly, patients with shorter disease duration (up to 5 years), had worse OCT results than healthy people, but the difference was not statistically significant.
- There is no mention in the discussion section about the authors' findings that there were no significant differences in OCT parameters between patients treated with various disease-modifying drugs.
Re.:
We mentioned in discussion about no significant differences in OCT parameters between patients treated with various disease-modifying drugs and we discuss this results:
In our study, the use of DMDs had no significant effect on the results of OCT. However, subgroups of patients treated with each drug were small and it could have influenced this observation. Current DMTs are effective in inhibiting neuroinflammation and reducing the rate of clinical relapses but their neuroprotective effects are less well defined. Data on the relationship between OCT results and DMDs are unclear. Knier et al. analyzed the association between OCT changes and DMT treatment in patients with MS [37]. The authors did not find any difference between patients with or without DMDs but they examined only subjects with first line drugs (interferon beta, glatiramer acetate, dimethyl fumarate, and teriflunomide). On the other hand, You at al. observed that progressive loss of retinal ganglion cells was more pronounced in patients treated with interferon beta than other DMTs (glatiramer acetate, fingolimod, dimethyl fumarate, natalizumab, alemtuzumab, rituximab, and ocrelizumab) [38]. Zivadinov et al. noticed that glatiramer acetate might have a neuroprotective effect against RNFL loss [39]. Button et al. found that GCIPL thinning in patients on both interferon beta and glatiramer acetate is faster than in patients on natalizumab [40].
- I do not entirely agree with the conclusion that the educational level influenced the results of OCT. Did the identified differences not result from the older age of people with primary education compared to those with higher education? In my opinion, the authors' conclusions regarding the dependence of OCT parameters on education are too far-reaching.
Re.:
We removed this conclusion from the discussion.
- I am surprised the authors did not mention the limitations of the study.
Re.:
We added Limitation section at the end of the paper:
Our study has some limitations. Firstly, the study group was small and came only from one centre. Secondly, although some researchers believe [31,41] that the analysis of OCT parameters in eyes without the history of ON is more reliable to illustrate the processes of neurodegeneration than in eyes with the history of ON, we assessed both eyes of each patient regardless of the history of ON. However, patients with ON in the last 6 months prior to the recruitment were excluded from the study. Thirdly, we did not evaluate the inner nuclear layer (INL), the parameter that recently has been dis-cussed as a marker of neurodegeneration in MS [42-45], so further study is needed in this field.
- Tables contain too much data, which makes them difficult to read. In table 8 the last parameter should be signed as GCIP + mRNFL instead of GCIPL +pRNFL.
Re.:
For better reading symbols “p=” were deleted, we used the leading zeroes for p-values and R’ coefficients (from interval from 0 to 1), e.g.: .25 insteed of 0.25. We used symbols * instead of p<0.05. Tables were simplified and corrected.

Reviewer 3 Report
In the present study, Torbus and colleagues demonstrated an association between OCT parameters and disability in two selected groups of MS patients divided for disease duration. The article agrees with previous literature that has shown the efficacy of OCT in identifying the neurodegenerative processes underlying the disease. Although the article may have scientific relevance, it needs several revisions:
- The authors found an association between OCT parameters and EDSS in the group of older patients. Please use logistic regression analysis to value the impact of disease duration and age on this correlation.
- The article is conducted on a small cohort of patients, and data obtained should be demonstrated on a larger population. Please underline the study's limitations and modify the phrase "our study demonstrates beyond doubt…" at line 406.
- Please add a limitations section in the discussion
- There are many tables in the study, and some data shown in the descriptive statistics do not find a correlation with the content of the article (for example, "family status"). Please simplify and reduce the tables.
Author Response
First, we would like to thank for this very detailed analysis of the manuscript. We agree with the majority of the Reviewers’ comments. Our replies are written in a red. We made several changes in our manuscript believing they address all the Reviewers’ comments (also marked in red).
Reviewer 3
In the present study, Torbus and colleagues demonstrated an association between OCT parameters and disability in two selected groups of MS patients divided for disease duration. The article agrees with previous literature that has shown the efficacy of OCT in identifying the neurodegenerative processes underlying the disease. Although the article may have scientific relevance, it needs several revisions:
- The authors found an association between OCT parameters and EDSS in the group of older patients. Please use logistic regression analysis to value the impact of disease duration and age on this correlation.
Re.:
Thank you for the comment. We have expanded the statistical analysis. Now the study takes into account the results of multiple regression.
Page 4, lines 165-167:
The multiple regression analysis were used to confirm a relationship between OCT parameters and patients’ age and disease duration.
Page 10, lines 310-313, Table 8:
Additional multiple regression analysis confirmed a significant influence of patients' age in the case of pRNFL Total, pRNFL Inferior, pRNFL Temporalis, and MS duration time in: pRNFL Total, mRNFL, GCIPL, GCIPL + mRNFL (Table 8).
- The article is conducted on a small cohort of patients, and data obtained should be demonstrated on a larger population. Please underline the study's limitations and modify the phrase "our study demonstrates beyond doubt…" at line 406.
Re.:
We added in Limitation section:
Our study has some limitations. Firstly, the study group was small and came only from one centre.
We modify the phrase "our study demonstrates beyond doubt…":
Our study confirmed that neuronal and axonal damage in MS progresses with the du-ration of the disease…
- Please add a limitations section in the discussion
Re.:
We added Limitation section at the end of the paper:
Our study has some limitations. Firstly, the study group was small and came only from one centre. Secondly, although some researchers believe [31,41] that the analysis of OCT parameters in eyes without the history of ON is more reliable to illustrate the processes of neurodegeneration than in eyes with the history of ON, we assessed both eyes of each patient regardless of the history of ON. However, patients with ON in the last 6 months prior to the recruitment were excluded from the study. Thirdly, we did not evaluate the inner nuclear layer (INL), the parameter that recently has been dis-cussed as a marker of neurodegeneration in MS [42-45], so further study is needed in this field.
- There are many tables in the study, and some data shown in the descriptive statistics do not find a correlation with the content of the article (for example, "family status"). Please simplify and reduce the tables.
Re.:
For better reading symbols “p=” were deleted, we used leading zeroes for p-values and R’ coefficients (from interval from 0 to 1), e.g.: .25 insteed of 0.25. We used symbols * instead of p<0.05. Tables were simplified.
Table 1 lists the characteristics of the variables that were used in the analysis (we removed some variables, like family status, children and employment).

Round 2
Reviewer 1 Report
The work is improved. I have no further comments.
Author Response
We would like to thank for this very detailed analysis of the manuscript and all the comments.
Reviewer 2 Report
The authors addressed all the remarks. I have no additional comments.
Author Response

(The authors gave the same response as above.)
